# A Dual-Branch Speech Enhancement Model with Harmonic Repair

**Lizhen Jia [1,2], Yanyan Xu [1,2,\*] and Dengfeng Ke [3]**

1   School of Information Science and Technology, Beijing Forestry University, Beijing 100083, China;
    jializhen@bjfu.edu.cn
2   Engineering Research Center for Forestry-Oriented Intelligent Information Processing of National Forestry
    and Grassland Administration, Beijing Forestry University, Beijing 100083, China
3   School of Information Science, Beijing Language and Culture University, Beijing 100083, China;
    dengfeng.ke@blcu.edu.cn
*   Correspondence: xuyanyan@bjfu.edu.cn

**Abstract:** Recent speech enhancement studies have mostly focused on completely separating noise from human voices. Due to the lack of specific structures for harmonic fitting in previous studies and the limitations of the traditional convolutional receptive field, there is an inevitable decline in the auditory quality of the enhanced speech, leading to a decrease in the performance of subsequent tasks such as speech recognition and speaker identification. To address these problems, this paper proposes a Harmonic Repair Large Frame enhancement model, called HRLF-Net, that uses a harmonic repair network for denoising, followed by a real-imaginary dual branch structure for restoration. This approach fully utilizes the harmonic overtones to match the original harmonic distribution of speech. In the subsequent branch process, it restores the speech to specifically optimize its auditory quality to the human ear. Experiments show that under HRLF-Net, the intelligibility and quality of speech are significantly improved, and harmonic information is effectively restored.

**Keywords:** speech enhancement; denoising; harmonic; fast Fourier convolution; dual branch structure

## 1. Introduction

In both real-world production and living scenarios, as well as modern communication devices, interference with audio signals is inevitable. Part of the interference originates directly from the real-world environments where voice information is collected, and part arises from signal degradation during compression, transmission, and sampling in electronic devices. This phenomenon is referred to as voice degradation. Speech enhancement technologies aim to remove background noise from audio as much as possible while retaining the original speech information. Traditional speech enhancement methods generally work based on statistical signal principles, such as spectral subtraction [1], minimum mean square error estimation [2], filtering methods including Wiener filter [3] and Kalman filter [4], and subspace enhancement methods that use cross-spectral pairs for frequency filtering of subspace signals [5]. However, these traditional methods often struggle to effectively reduce noise, especially in the presence of multiple noise sources or when the noise frequency range is concentrated. To devise a more versatile filtering method, S. P. Talebi proposes an approach based on fractional calculus [6], aiming to address setting $\alpha$-stable statistics more effectively, which provides an alternative solution to the requirements of modern filtering applications.

Currently, the mainstream speech enhancement methods based on deep learning follow two technical approaches. One is time-domain-based, utilizing neural networks to directly infer the spectrum of pure speech from noisy speech, which may produce better harmonic results but require more computational resources and may be less effective in suppressing non-stationary noise compared to time-frequency domain methods [7]. Time-domain methods for waveform processing can significantly improve the Signal-to-Distortion Ratio (SDR) [8], but they may lead to a decrease in auditory perception. The

primary reason for this issue is that the system, working in only one transform domain, struggles to filter out redundant information in the background noise. The other approach is frequency-domain-based, typically using masking techniques. The basic idea is to combine speech and noise signals in a certain way so that the predicted mask can accurately separate speech and noise signals. Complex Ideal Ratio Mask (CIRM), based on Fourier transformation and the concept of a crude ideal ratio mask, not only considers amplitude information but also phase information to preserve the phase information of the original speech signal and avoid signal distortion due to amplitude changes [9].

Previous research often underestimates the importance of phase information in speech repair, leading to unavoidable speech distortion in denoised speech, which significantly interferes with subsequent speech recognition and speaker recognition tasks, reducing their performances. Hu et al. demonstrate [10] that better utilization of phase information in speech signals can significantly improve the quality of enhanced speech, achieving better performances with less loss. Direct estimation of phase information in spectrograms is challenging, often resulting in large neural networks [11]. To allow the phase information of the speech to play a greater role in the denoising process, researchers have made a considerable amount of effort. Inspired by the Taylor series, Li [12] and others propose a decoupled speech enhancement framework, dividing the optimization problem of the complex spectrum into two parts: the optimization problem of the magnitude spectrum and the estimation of complex residues. To refine the phase distribution, they define the difference between the rough spectrum and the target spectrum to measure the phase gap. A dual-branch enhancement network is introduced in [13], where the complex spectrum refinement branch collaboratively estimates the amplitude and phase information of speech by taking in both the real and imaginary parts. In the work of [14], a dedicated path encoder-decoder is designed to restore phase information and generate the phase spectrum for predicting speech. Experimental results have shown that the neural network's receptive field significantly affects the efficiency of model parameter utilization. Therefore, expanding the model's receptive field to be more sensitive to contextual information can achieve better phase understanding. Additionally, processing amplitude and phase spectra information as separate branches in neural networks can better utilize phase information in speech signals, offering better interpretability.

Therefore, in this paper, we propose the harmonic repair large frame enhancement model, HRLF-Net, which is a dual-branch speech enhancement model designed with specialized modules to predict the harmonic distribution of speech. In the real-part branch of the network, we utilize fast Fourier convolutional operators instead of traditional 2D convolutions for amplitude spectrum repair, which effectively expands the model's receptive field and significantly improves the performance of speech harmonics. An architecture with dilated DenseNet and deconvolution blocks is deployed in the imaginary branch to fully utilize speech phase information while preserving the temporal characteristics of the speech signal, making the enhanced speech more accurately reflect the dynamic changes of the original speech. HRLF-Net is tested on two public datasets, VoiceBank + DEMAND [15] and DNS Challenge 2020 [16]. Experimental results show that it outperforms most existing models and achieves state-of-the-art results in terms of Perceptual Evaluation of Speech Quality (PESQ).

## 2. Proposed Methods

This article primarily addresses the issue of standard single-channel speech enhancement, aiming to construct a neural network whose target is to fit CIRM that transforms waveforms with additive noise into pure speech waveforms. The following sections will provide detailed descriptions of the key components and the overall composition of the model.

### 2.1. Fast Fourier Convolution (FFC)

The Fast Fourier Transform (FFT) converts time-domain signals into frequency-domain signals. Compared to the Short-Time Fourier Transform (STFT), which decomposes time-domain signals into spectral components of a series of window functions, FFT first decomposes the signal into a sum of sine and cosine functions to represent the spectrum, thus efficiently computing the Discrete Fourier Transform and obtaining the spectral information of the signal.

For traditional fully convolutional models, the growth of the effective receptive field is too slow, and the lack of an effective context-capturing structure often results in suboptimal enhancement effects, a problem that is more prominent in wideband, long-duration audio. In the amplitude spectrogram of speech, harmonic structures often form periodic patterns, a feature that is suitable for processing with FFC, i.e., repetitive microstructures. Considering the model's aim to expand the neural network's receptive field for speech context, using FFC is more appropriate for analyzing the entire speech spectrum. It is a widely used non-global operator in the field of Computer Vision (CV) and can replace traditional convolution layers in network architectures, playing a significant role in repairing damaged periodic backgrounds.

In computer vision research, the Fourier transform generally applies a complex two-stage method. In this work, we set the working domain of the Fourier transform as the frequency component of the feature map. The basic structure of FFC is shown in Figure 1. Specifically, the basic structure of FFC is implemented as follows.

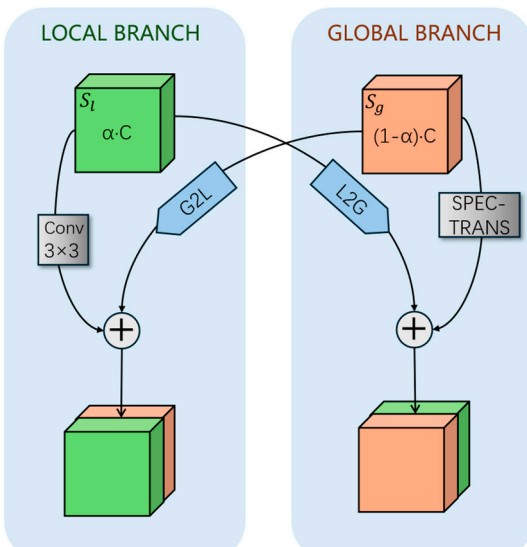

**Figure 1.** The basic structure of FFC.

- Before the signal S enters the operator, it is divided along the feature channels of the feature map into the local block $S_l$ and the global block Sg, where the local block $S_l$ will use adjacent local blocks as learning objects, and the remaining global block Sg is used to obtain speech context associations. We use parameter $\alpha$ to control the division ratio of channels.
- To cover the entire spectrum with the receptive field of the global block, the original feature space is transformed into a specific domain on the global Fourier unit, and after the spectral data is updated, it is restored to a spatial format. Meanwhile, additional segmentation and combination are performed in the local Fourier unit to make it more sensitive to spectral detail features. Finally, the output data of the two units are connected using a residual connection.
- The results of the global and local blocks are simply connected to form the output of the complete operator. At this point, the entire FFC module is fully differentiable and can replace all traditional convolutions.

We apply a real one-dimensional fast Fourier transform on the frequency dimension of the input feature map at the global component level and then concatenate the real and imaginary parts of the spectrum along the channel dimension. Next, we apply convolutional blocks on the frequency domain and finally restore the spatial structure using inverse FFT.

### 2.2. The Harmonic Repair Module

Although speech signals can be extensively damaged due to noise, the harmonic parts usually reside in higher energy regions and are not completely masked. Since deep learning models prioritize fitting high-energy and more robust (prominent) harmonic structures due to gradient descent and convergence [17], harmonic waveforms exhibit significant comb-like features, meaning that even if part of them are damaged by noise, the remaining parts contain information that can infer the original harmonic distribution. To model harmonic data in the spectrum, the model uses a harmonic-to-fundamental frequency transformation matrix $Q$ [18], which calculates the corresponding harmonic distribution using the predicted fundamental frequency. The input $X_P \in \mathbb{R}^{T \times F}$, after convolution energy normalization, produces a query-key matrix $K$. Matrix multiplication between $K$ and $Q$ and the application of the sigmoid method obtains a confidence vector for the pitch of the fundamental frequency, indicating the likelihood of each candidate value corresponding to the pitch.

The harmonic repair module uses high-resolution comb tooth spacing to infer the damaged harmonic distribution and fine-tunes the result using convolution. Figure 2 shows the structure of the harmonic repair module. Unlike the traditional attention mechanism, which calculates attention weights using query vectors and key-value pairs, the harmonic repair mechanism calculates and repairs harmonic information based on the spectral and harmonic-pitch converter, using a residual connection [19] to mitigate the vanishing gradient problem. The locality of convolution is the fundamental guarantee for harmonic modeling, so the module retains the spectral structure even after processing [20].

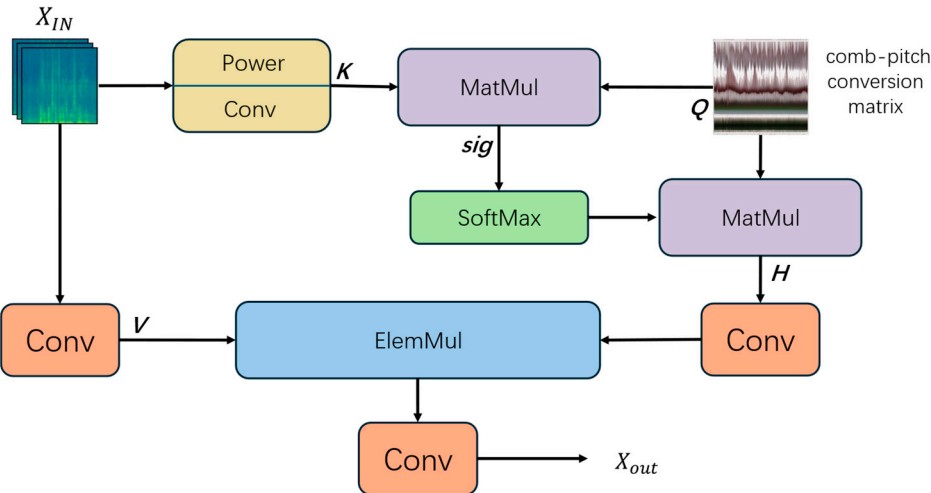

**Figure 2.** The structure of the harmonic repair module.

The harmonic distribution $H$ can be represented as $softmax\left[sig\left(K \cdot Q^T\right)\right]$. $X_P$ and $H$ are rectified using one-dimensional convolution, and then $H$ is applied to $X_P$ for element-wise multiplication, and the result is convolved to output $X_{out}$. Using the harmonic repair module can effectively repair and restore voice-focused frequency bands in audio and suppress the impact of harmonic-like noise in noisy speech on the enhancement results.

### 2.3. The Harmonic Fading-Out Module

Due to the absence of channel interaction in the harmonic repair module, there is redundancy in the restoration process of mid-to-high frequency harmonics, negatively

impacting the fidelity of speech timbre restoration. The harmonic fading-out module extracts spectral information from multiple angles and filters out potentially over-restored comb-like waveforms. It connects the harmonic repair module to two stacked multi-head attention [21] modules, one unfolding along the channel dimension and the other along the frequency dimension.

As illustrated in Figure 3, this module first reshapes the input data $X$ into $\mathbb{R}^{C \times F \times L}$ and uses three linear layers to obtain the $Q$, $K$, and $V$ keys. $L$ and $F$, respectively, represent the number of time frames and frequency bins. After rectification via Scale, $Qc$ and $Kc$ are multiplied and then combined with $Vc$, and finally, the result is concatenated with $Xc$ to form a residual connection. The output is then used as the input for the next layer of the frequency attention module. The frequency attention also operates in a similar structure, ultimately producing the output of this module $X \in \mathbb{R}^{C \times L \times F}$.

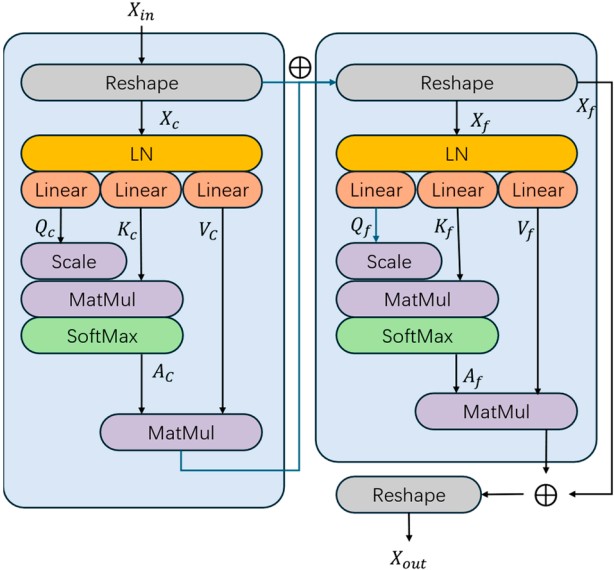

**Figure 3.** The structure of the harmonic fading-out module.

### 2.4. Time Series Modeling

To avoid the issues caused by the 3S paradigm adopted in the recent Continuous Speech Separation (CSS) systems [22–24], such as the increased computational burden due to multiple overlaps between windows, and the dilemma of choosing window length for the performance and stitching stability, for time series modeling, the enhancement network employs Long Short-Term Memory (LSTM) layers with memory skipping to capture the contextual information of speech [25]. This layer is an improvement on LSTM [26], and the traditional LSTM mapping function can be represented as

$$\hat{W}, \hat{c}, \hat{h} = LSTM(W, c, h), \tag{1}$$

where $W \in \mathbb{R}^{T \times N}$ is the input sequence, and $c$ and $h$ are the initialized cell state and hidden state, respectively. On the left side of the equation, $\hat{W}$ represents the output sequence, while $\hat{c}$ and $\hat{h}$ are the updated cell state and updated hidden state, respectively. Generally, it is believed that $\hat{c}$ encodes the entire long-term memory sequence to form long-term memory, while $\hat{h}$ is used for short-term memory of the processed sequence.

In the time domain task of speech, T in the input sequence $W \in \mathbb{R}^{T \times N}$ can usually take a very large value. $W$ can be divided into several smaller segments $[w_l^1, w_l^2, \cdots, w_l^S]$, where $S$ is the total number of segments, and the length of each segment is determined using the parameter $K$, with $W_l^s = W[sK - K : sk, :] \in \mathbb{R}^{K \times N}$.

The Skipping-Memory LSTM, which consists of L basic layers connected in series, is shown in Figure 4. In each layer structure, $S$ seg-LSTMs are used to process the $S$ small

segments $\left[ w_l^1, w_l^2, \cdots, w_l^S \right]$, where l represents the l-th input. At this point, the mapping function of the seg-LSTM can be expressed as

$$\overline{W}_{l+1}^S, c_{l+1}^S, h_{l+1}^S = \text{Seg\_LSTM}\left( W_l^s, \hat{c}_l^s, \hat{h}_l^s \right) \tag{2}$$

and

$$W_{l+1}^s = \text{LN}\left( \overline{W}_{l+1}^s \right) + W_l^s \tag{3}$$

where LN is the layer normalization operation in the residual connection, and $\hat{c}_1^s$, $\hat{h}_1^s$ are all initialized to 0. All the segments are finally collected in the Mem-layer for global modeling to conduct cross-segment processing. The memory-skipping LSTM constructed in this way can handle relatively long speech sequences. By purposefully discarding the overlapping areas of adjacent segments, it achieves an effective balance between performance and efficiency.

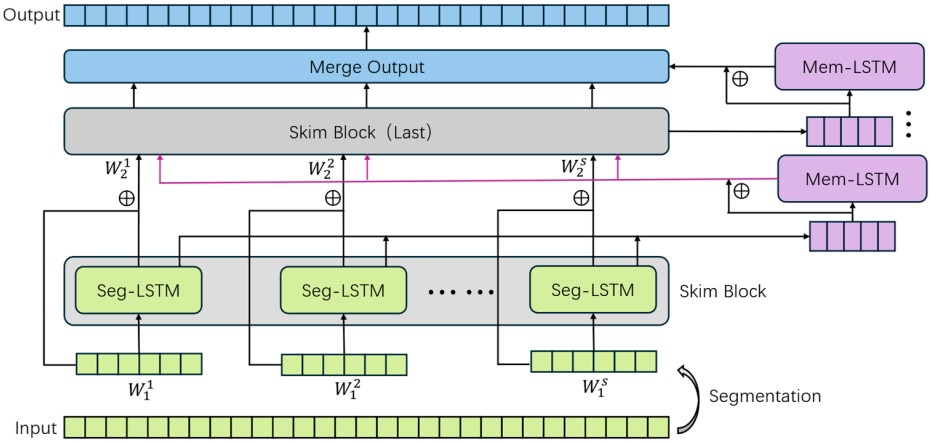

**Figure 4.** LSTM with skip connections.

### 2.5. The Harmonic Repair Large Frame Enhancement Model HRLF-Net

Under normal circumstances, the noise in noisy speech is non-additive, and the noisy speech $y \in \mathbb{R}^{p \times l}$ can be represented as $y = s + n$, where $s$ is the pure speech signal desired in the task, $n$ is the noise, and $p$ is the total number of samples in the signal. The task of speech enhancement is to separate $s$ from $y$ as much as possible while suppressing the noise as much as possible.

The input to the proposed model is the complex spectrum after STFT, represented as $Y = S + N$, where Y, S and N respectively represent the complex spectra of the noisy speech, pure speech, and noise signal, with $\{Y, S, N\} \in \mathbb{R}^{2T \times F}$. T and F represent the number of bins in the time and frequency dimensions in the real and imaginary parts, respectively. Figure 5 shows the overall architecture of our proposed HRLF-Net.

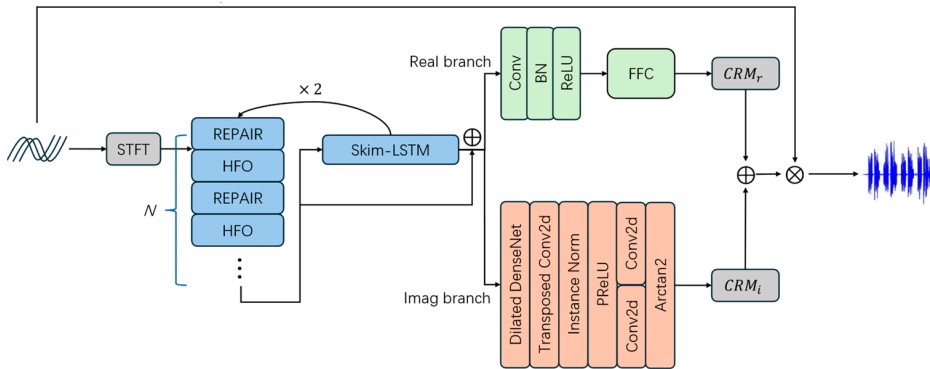

**Figure 5.** The overall structure of HRLF-Net.

Several harmonic repair modules are used to extract and refine features, and harmonic fading-out modules, represented as HFO in Figure 5, are applied after each processing to prune the processed features to a certain extent. Harmonic fading-out modules are used concurrently with each harmonic repair module for optimal filtering.

Real and Imag branches correspond to the real and imaginary parts of the CIRM mask structure and are processed using two branch networks. In the real branch, convolutional layers with causal 2D convolution, batch normalization [27], and PReLU [28] preliminarily filter the input signal and separate different channels, which is an autoencoder design [29], followed by cascaded FFC modules forming residual connections. The resulting output is up sampled to obtain the predicted CIRM spectrum's real part output $M_r$. To avoid issues of non-structural and wrapping phase jumps [30], the Imag branch employs a phase decoder with dilated DenseNet [31]. After the deconvolution block, parallel dual 2D convolution layers output pseudo-components, and a dual-parameter arctangent function activates these two components to obtain the predicted CIRM imaginary spectrum output $M_i$, where instance normalization layers are connected to standardize the network's intermediate features. This structure is based on the definition of the complex ideal ratio mask.

$$M_r = \frac{Y_r S_r + Y_i S_i}{Y_r^2 + Y_i^2} \tag{4}$$

and

$$M_i = \frac{Y_r S_i + Y_i S_i}{Y_r^2 + Y_i^2}, \tag{5}$$

where $Y_r$ and $Y_i$ respectively represent the real and imaginary parts of the noisy complex spectrum. $S_r$ and $S_i$ represent the real and imaginary parts of the pure speech complex spectrum. The masks calculated using $M_r$ and $M_i$, estimated from the noisy speech, can be represented using the following formula:

$$M = \frac{Y_r S_r + Y_i S_i}{Y_r^2 + Y_i^2} + i\frac{Y_r S_i + Y_i S_i}{Y_r^2 + Y_i^2}, \tag{6}$$

Multiplying the noisy speech spectrum $Y = Y_r + Y_i$ with the mask $M = M_\gamma + jM_i$ results in the enhanced speech spectrum, which is then transformed back into the final waveform (W) using the Inverse Short-Time Fourier Transform (ISTFT):

$$\widetilde{w} = Y_\gamma M_r - Y_i M_i + \mathrm{i}(Y_r M_i + X_i M_r). \tag{7}$$

*2.6. Loss Functions*

We use a multilayer loss function to help the network effectively train and fit [28]. The time loss $L_T$ is calculated by computing the L1 norm difference between the clean speech waveform and the model's output-enhanced waveform $\mathbb{E}_{x,\hat{x}}[\|x - \hat{x}\|_1]$, with the time loss result being the average of the differences across all frames. The magnitude loss $L_M$ is computed by first extracting the magnitude spectra $X_m$ and $\overline{X}_m$ from the clean speech and the enhanced speech, respectively, and then calculating the Mean Squared Error (MSE) between them $\mathbb{E}_{X_m,\hat{X}_m}\left[\|X_m - \hat{X}_m\|_2^2\right]$. The final loss value is the expected MSE across all frequency and time points. The function composition also includes the complex spectrum loss $L_C$, which uses the ground truth and predicted complex spectra as inputs and calculates the MSE between the two spectral graphs $\mathbb{E}_{X_r,\hat{X}_r}\left[\|X_r - \hat{X}_r\|_2^2\right] + \mathbb{E}_{X_i,\hat{X}_i}\left[\|X_i - \hat{X}_i\|_2^2\right]$, yielding the expected errors of the real and imaginary parts as the final output of $L_C$.

The polynomial of the above losses is used as the final loss function, and appropriate hyperparameters are set to optimize the model's performance.

## 3. Experiments

In this section, the datasets and performance evaluation metrics used in our experiments are firstly introduced. Then, the effectiveness of the proposed modules is validated in subsequent experiments, demonstrating overall performance superiority across several metrics compared with a series of strong baselines.

### 3.1. Datasets and Experimental Setup

**DNS Challenge:** To maintain consistency with the evaluation datasets of mainstream models, in our experiments, we utilize the ICASSP 2020 DEEP NOISE SUPPRESSION CHALLENGE (DNS Challenge 2020) dataset for training and evaluation. The dataset includes clean speech predominantly in English, from 2150 speakers selected from over ten thousand individuals, totaling over 500 h, as well as 60,000 noise segments from 150 different noise categories. During model training, clean speech and noise are randomly selected from their respective sets and dynamically combined to simulate noisy speech. The Signal-to-Noise Ratio (SNR) is uniformly distributed between $-5$ dB and 20 dB. The training and validation sets are split in a 4:1 ratio and are completely isolated from the test set.

**VoiceBank + DEMAND:** This public dataset includes paired noisy and clean speech clips. The clean speech audio segments are sourced from the corpus, which contains 28 different speakers, 11,518 audio segments, and over 800 speech segments for the test set. Clean speech segments are mixed with ten different types of noise at SNRs of 0 dB, 5 dB, 10 dB, and 15 dB. The test set includes DEMAND database materials at SNRs of 2.5 dB, 7.5 dB, 12.5 dB, and 17.5 dB.

The model input uses a Hanning window with a length of 20 ms and 50% overlap, along with an STFT of 320 points, to produce 161-dimensional spectral features. The size and stride of the convolutional kernel are (2,3) and (1,1), respectively, and the number of heads in the harmonic integration of the harmonic repair is set to 4. The channel numbers of the harmonic repair module are {12, 24, 24, 48, 48, 24, 12, 12}, where the first six are for the main structure, and the last two belong to the compensation components. The SkiM layer includes four basic SkiM blocks, with the hidden dimension of the LSTM set to 256. The feature-length segment size S for Seg-LSTM is set to 150, where only layer normalization is performed in the feature dimension in the causal SkiM. In the FFC module, the channel ratio used in the global branch is $\alpha$, which equals to 0.75. The imaginary part branch applies four convolutional layers with dilation sizes of 1, 2, 4, and 8 in the expanded DenseNet sequentially. The model undergoes 80,000 iterations of training, with a batch size of eight. The Adam optimizer is employed, and the learning rate is 0.0001.

### 3.2. Evaluation Metrics

PESQ: Perceptual Evaluation of Speech Quality is a standardized objective metric used to assess the quality of speech signals. Initially developed by the International Telecommunication Union (ITU), it was primarily designed for evaluating speech quality within telephony systems. PESQ aims to provide an objective quantification of speech quality by simulating aspects of the human auditory system's response to audio signals.

STOI: Short-Time Objective Intelligibility is an objective metric designed to assess the intelligibility of speech signals. Similar to PESQ, STOI aims to provide an objective and quantitative method for measuring the understandability of speech signals. This metric primarily focuses on the intelligibility of speech signals in noisy environments and is applicable to speech communication, speech recognition, and other applications that involve conveying speech information in the presence of background noise.

SI-SDR: Scale-Invariant Signal-to-Distortion Ratio is an objective metric used to measure the quality of separating audio sources. It is primarily employed to assess the performance of separation algorithms in audio source separation tasks. SI-SDR considers a balance between the scale-invariant proportion between the estimated and true source signals and the level of distortion, making it a scale-invariant measure.

CSIG: Composite Speech Intelligibility Index is a metric used to assess the performance of speech processing systems, such as speech enhancement and speech coding. This metric is primarily employed to measure the intelligibility and quality of speech signals. The design of CSIG aims to provide a comprehensive evaluation that reflects the impact of speech processing systems on the original speech signal. It considers not only the clarity of the speech signal but also the combined effects of factors such as background noise and distortion.

### 3.3. Parameter Selection

Table 1 shows the experiments conducted on the DNS Challenge 2020 dataset for selecting a suitable parameter $N$, which is the number of paired Repair and HFO blocks. From Table 1, we see that as $N$ starts to increase from 2, all metrics in the table show a significant improvement, with PESQ being the most obvious. When $N$ exceeds 4, STOI does not show a noticeable increase, while other metrics begin to decline to different extents and contribute to an increase in the model parameters. Based on these results, in our following experiments, we set $N$ to be 4.

**Table 1.** Selection of the number of stacked blocks for Repair and HFO Blocks.

| Number of Blocks ($N$) | PESQ | STOI (%) | SI-SDR | CSIG | Para. (M) |
|:---:|:---:|:---:|:---:|:---:|:---:|
| 2 | 3.211 | 96.43 | 20.13 | 4.15 | 11.943 |
| 3 | 3.350 | 97.54 | **20.64** | 4.08 | 12.503 |
| 4 | **3.513** | **97.85** | 20.52 | **4.22** | 13.105 |
| 5 | 3.498 | 97.76 | 20.55 | 4.12 | 13.842 |

The optimal values of the objective evaluation metrics are highlighted in bold font.

### 3.4. Ablation Studies

This study, based on ablation experiments, analyzes the necessity of performance improvement for each module of the model network. It then evaluates the overall performance of the model based on PESQ, STOI, and SI-SDR. Finally, the comprehensive performance of the proposed method is verified. The ablation experiments are conducted on the DNS Challenge 2020 dataset and the VoiceBank + DEMAND dataset.

The experiment results in Tables 2 and 3 show that using only the harmonic repair module and disabling the fading-out module leads to a significant decrease in speech quality. This is because the nonlinear operations of the harmonic repair module cause disturbances, especially in the recovery of harmonics and particularly in high-amplitude sections, amplifying these disturbances and ultimately affecting the enhancement performance. Pairing the fading-out module with harmonic repair significantly mitigates this issue. Replacing SkiM with naive LSTM demonstrates that SkiM achieves essentially the same performance level as naive LSTM while significantly reducing computational costs. Comparative experiments using the original 2D convolutional kernels for the real part branch's FFC module are conducted. These experiments show that thanks to FFC's advantage of having a large-scale receptive field, the context of speech is fully utilized in the recovery process of the real spectrogram. Unlike traditional convolution, where the receptive field is limited by the size of the kernel, FFC operates convolution in the frequency domain, considering all frequency components of the input simultaneously, which allows for better capture of long-term dependencies in the signal. This demonstrates the advantage of FFC's large-scale receptive field in fully utilizing the context of speech in the recovery process of the real spectrogram.

**Table 2.** Ablation experiments targeting the main submodules on the DNS Challenge 2020 dataset.

| Model | Para. (M) | PESQ | STOI (%) | SI-SDR |
|---|---|---|---|---|
| **HRLF-Net** | 13.105 | 3.513 | 96.68 | 20.36 |
| -Fading-out | 12.245 | 3.229 | 96.26 | 17.50 |
| -Harmonic repair | 9.535 | 3.326 | 95.88 | 16.38 |
| -skimLSTM | 7.630 | 3.175 | 95.24 | 15.92 |
| +LSTM | 9.36 | 3.198 | 95.56 | 16.42 |
| -FFC | 12.805 | 3.372 | 95.22 | 17.15 |
| +Conv2D (origin) | 13.065 | 3.397 | 96.13 | 17.39 |

To serve as a control, the effectiveness of FFC is studied using conventional 2D convolution substitution, where +/- indicates whether the submodule is masked in the experiments.

**Table 3.** Ablation experiments targeting the main submodules on the VoiceBank + DEMAND dataset.

| Model | Para. (M) | PESQ | STOI (%) | SI-SDR |
|---|---|---|---|---|
| **HRLF-Net** | 13.105 | 3.519 | 96.78 | 18.67 |
| -Fading-out | 12.245 | 3.329 | 96.36 | 17.50 |
| -Harmonic repair | 9.535 | 3.426 | 95.98 | 16.38 |
| -skimLSTM | 7.630 | 3.275 | 95.34 | 15.92 |
| +LSTM | 9.36 | 3.298 | 95.66 | 16.42 |
| -FFC | 12.805 | 3.472 | 95.32 | 17.15 |
| +Conv2D (origin) | 13.065 | 3.477 | 96.23 | 17.39 |

To serve as a control, the effectiveness of FFC is studied using conventional 2D convolution substitution, where +/- indicates whether the submodule is masked in the experiments.

### 3.5. Comparison with Other Models

We compare the proposed model with other models on two datasets using the experimental results provided in the original papers. On the DNS Challenge dataset, as seen in Table 4, our model improved PESQ without significantly increasing the number of model parameters and maintained an advanced level in other objective metrics. On the Voicebank + DEMAND dataset, as shown in Table 5, except for slight inferiority to DEMUCS [30] in CSIG, our model similarly shows significant improvements in all metrics. Compared to other studies, the advantage of our proposed method lies in considering the crucial role of harmonic information under noise masking for speech restoration. When estimating the phase spectrum, we follow the complex spectrum calculation method separately from the real and imaginary parts. We achieve collaborative optimization of the phase spectrum and magnitude spectrum using multiple losses while also considering contextual information in the speech.

**Table 4.** Comparison with other models on the DNS Challenge 2020 dataset.

| Model | PESQ | STOI (%) | SI-SDR | CSIG | #Params. (M) |
|---|---|---|---|---|---|
| 2020 DCCRN [10] | 2.711 | 96.0 | 17.967 | 2.98 * | 3.67 |
| 2020 DEMUCS [32] | 2.20 | 82.0 | 15.56 | 3.44 * | 33.5 |
| 2022 Fullsubnet-plus [33] | 2.487 | 79.6 | 18.34 | 3.12 * | 8.6 |
| 2022 TaylorSENet [12] | 3.23 | 97.69 | 19.78 | 3.49 * | 5.4 |
| 2023 MP-SENet [34] | 3.50 | **98.00** | 20.31 | **4.73 *** | 13.24 * |
| Ours **HRLF-Net** | **3.513** | 96.68 | **20.36** | 4.22 * | 13.105 |

* indicates that the corresponding result is not provided in the original paper, and the value in this table is obtained through our experiments.

**Table 5.** Comparison with other models on the VoiceBank + DEMAND dataset.

| Model | PESQ | STOI (%) | SI-SDR | CSIG | #Params. (M) |
|---|---|---|---|---|---|
| 2020 DEMUCS [10] | 3.03 | 87.0 * | 18.5 * | 4.36 | 60.8 |
| 2022 HiFi++ [35] | 2.76 | 89.53 * | 18.4 * | 4.09 | 1.7 |
| 2022 CMGAN [36] | 3.41 | 96.0 * | 16.34 * | **4.63** | 1.83 |
| Ours **HRLF-Net** | **3.519** | **96.78** * | **18.67** * | 4.23 | 13.105 |

\* indicates that the corresponding result is not provided in the original paper, and the value in this table is obtained through our experiments.

### 3.6. Spectrogram Analysis

Figure 6 displays the spectrograms before and after enhancement via our proposed model. The spectrogram comes from a randomly selected ten-second audio in the DNS Challenge 2020 set. The red rectangular boxes in the image highlight the pure noise sections that are almost devoid of speech. In these sections, the proposed model effectively suppresses the background noise, as can be seen in the comparative spectrograms below. The pure noise sections, containing no human voice, are almost completely silent after noise reduction. The white and black dashed boxes in this figure indicate the effectiveness of harmonic restoration. In the white dashed box, the harmonic features are nearly invisible due to noise masking. After enhancement, the corresponding section in the spectrogram shows distinct comb-like harmonics, which indicates that the proposed model significantly restores the harmonics of speech during the enhancement process. This plays a key role in improving the clarity and distinguishability of the speech.

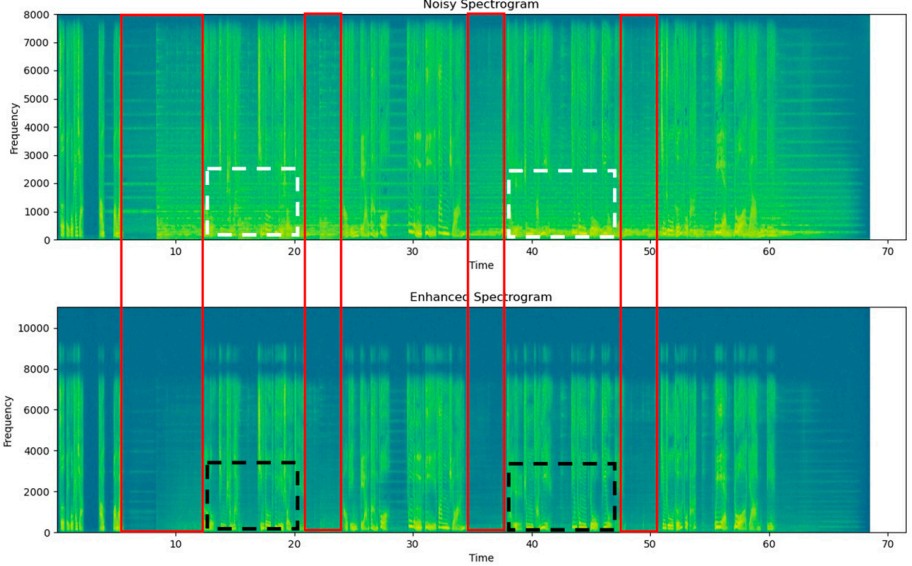

**Figure 6.** Comparison of spectrograms.

### 4. Conclusions

In this paper, we proposed HRLF-Net, a dual-branch speech-enhanced network. The harmonic repair module used in the model significantly restores the harmonic distribution of the speech. In the subsequent imaginary and real dual-branch structure, the FFC module plays a key role in expanding the receptive field of the model, while the dilated DenseNet effectively overcomes the phase wrapping and significantly improves the comprehensibility of the speech. The effectiveness and necessity of each module in the model are validated through ablation experiments. The proposed network shows a significant improvement in the PESQ metric compared to other state-of-the-art models, and it maintains a high level of short-term speech intelligibility.

Furthermore, during the experiments, we observed significant variations in the dynamic range of speech signals due to differences in speakers, contexts, and recording

conditions. Similarly, background noise ranges from low-intensity environmental noise to high-intensity interference signals such as traffic or industrial noise. Therefore, to further enhance the speech enhancement system's ability to handle dynamic range, we will explore additional data-driven approaches in the future. Also, we plan to extend our work to real-time applications by achieving even lower latency levels.

**Author Contributions:** L.J.: Conceptualization of this study, Methodology, Software, Writing. Y.X.: Supervision. D.K.: Writing. All authors have read and agreed to the published version of the manuscript.

**Funding:** This research received no external funding.

**Institutional Review Board Statement:** Not applicable.

**Informed Consent Statement:** Not applicable.

**Data Availability Statement:** The data presented in this study are openly available.

**Conflicts of Interest:** The authors declare that they have no known competing financial interests or personal relationships that could have appeared to influence the work reported in this paper.

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
