# Peer review of "A Dual-Branch Speech Enhancement Model with Harmonic Repair"

_applsci, doi:10.3390/app14041645_

Round 1

Reviewer 1 Report

Comments and Suggestions for Authors

Avery interesting manuscript with some minor required changes. Please see the attached file.

Author Response

 Please refer the attached file.

Reviewer 2 Report

Comments and Suggestions for Authors

The manuscript shows that we can use a harmonic repair network for denoising. The main subject of the article fits well within the journal's scope. 

Shortcomings:

1. All abbreviations should be explained at the first occurrence, even in the abstract section.  

2. The background research description is good but short and does not show different ways to achieve the solutions to the main manuscript issue. The background research should extend in these different ways. The conclusion section should be developed again after the description of the background research extension is grown.

3. After the section titles should be a short introduction before the subsection title.

4. References to figures in the text should appear before the place where the indicated figure is.

5. Keywords should be keywords, not sentences.

Comments on the Quality of English Language

 Language is good, but many misplaced words make the text hard to read.

Author Response

 Please refer the attached file.

Reviewer 3 Report

Comments and Suggestions for Authors

After reviewing the manuscript titled "A Dual Branch Speech Enhancement Model with Harmonic Repair," I have identified several key areas for improvement. This innovative study proposes a new approach to speech enhancement by focusing on harmonic repair and employing a dual-branch model structure. However, there are aspects that could be enhanced for greater clarity, depth, and overall scholarly contribution. Here are my detailed comments and suggestions:

  1. Clarity in Methodology: The methodology section outlines the novel approach of using fast Fourier convolution and a harmonic repair module. However, it would be beneficial to provide more detailed explanations about the specific algorithms used and their implementation in the model.

  2. Experimental Design and Data Analysis: The experiments conducted to validate the model are crucial. More detail about the experimental setup, including the datasets used, the conditions under which the experiments were conducted, and the rationale behind the choice of evaluation metrics, would strengthen this section.

  3. Results and Discussion: While the results indicate the model's effectiveness, a deeper analysis comparing these results to existing speech enhancement methods would add significant value. Discussing the model's performance in various noise conditions and its limitations would provide a more comprehensive understanding of its capabilities and applications.

  4. Practical Implications and Future Research Directions: The conclusion effectively summarizes the findings but could be expanded to discuss the practical implications of this technology in real-world applications. Additionally, suggesting areas for future research, particularly in terms of scalability and application in different acoustic environments, would be highly beneficial.

  5. Figures and Tables Quality: Ensure that all figures and tables are of high quality, clearly labeled, and directly referenced in the text. This will help in better illustrating and supporting your findings.

By addressing these points, the quality and impact of your manuscript can be significantly enhanced. The research holds promising potential for advancements in speech enhancement techniques, and I look forward to seeing the refined version of your work.

Author Response

 Please refer the attached file.

Round 2

Reviewer 3 Report

Comments and Suggestions for Authors

Good deal for paper